# Validation of a Harmonised, Three-Item Cognitive Screening Instrument for the Survey of Health, Ageing and Retirement in Europe (SHARE-Cog)

**DOI:** 10.3390/ijerph20196869

**Published:** 2023-09-30

**Authors:** Mark R. O’Donovan, Nicola Cornally, Rónán O’Caoimh

**Affiliations:** 1Health Research Board Clinical Research Facility, University College Cork, Mercy University Hospital, T12WE28 Cork, Ireland; markodonovan@ucc.ie; 2Catherine McAuley School of Nursing and Midwifery, University College Cork, T12AK54 Cork, Ireland; n.cornally@ucc.ie; 3Department of Geriatric Medicine, Mercy University Hospital, T12WE28 Cork, Ireland

**Keywords:** cognition, dementia, mild cognitive impairment, screening, SHARE

## Abstract

More accurate and standardised screening and assessment instruments are needed for studies to better understand the epidemiology of mild cognitive impairment (MCI) and dementia in Europe. The Survey of Health, Ageing and Retirement in Europe (SHARE) does not have a harmonised multi-domain cognitive test available. The current study proposes and validates a new instrument, the SHARE cognitive instrument (SHARE-Cog), for this large European longitudinal cohort. Three cognitive domains/sub-tests were available across all main waves of the SHARE and incorporated into SHARE-Cog; these included 10-word registration, verbal fluency (animal naming) and 10-word recall. Subtests were weighted using regression analysis. Diagnostic accuracy was assessed from the area under the curve (AUC) of receiver operating characteristic curves. Diagnostic categories included normal cognition (NC), subjective memory complaints (SMC), MCI and dementia. A total of 20,752 participants were included from wave 8, with a mean age of 75 years; 55% were female. A 45-point SHARE-Cog was developed and validated and had excellent diagnostic accuracy for identifying dementia (AUC = 0.91); very good diagnostic accuracy for cognitive impairment (MCI + dementia), (AUC = 0.81); and good diagnostic accuracy for distinguishing MCI from dementia (AUC = 0.76) and MCI from SMC + NC (AUC = 0.77). SHARE-Cog is a new, short cognitive screening instrument developed and validated to assess cognition in the SHARE. In this cross-sectional analysis, it has good–excellent diagnostic accuracy for identifying cognitive impairment in this wave of SHARE, but further study is required to confirm this in previous waves and over time.

## 1. Introduction

The world’s population is ageing such that the number of people aged ≥65 years of age is expected to more than double from 771 million in 2022 to 1.6 billion by 2050 [1]. This trend is associated with an increased prevalence of cognitive impairment, including dementia [2]. Estimates from the Global Burden of Disease study suggest that the number living with dementia will increase from 57.4 million in 2019 to 152.8 million by 2050 [3]. Ageing is also associated with other cognitive issues, which herald or predispose to the development of dementia, such as mild cognitive impairment (MCI), a minor neurocognitive disorder characterised by symptoms and observable deficits in cognition without a loss of function [4], as well as subjective memory complaints (SMC) [5].

For healthcare practitioners, the early detection of cognitive symptoms may improve outcomes by facilitating the prompt initiation of multi-modal management strategies including lifestyle risk factor modification and treatment with cholinesterase inhibitors for those with dementia [6,7]. Recent studies showing the potential for disease modifying therapies have added impetus to this. Although the evidence for population or community-level cognitive screening in asymptomatic individuals is limited and has little support at present [8,9,10], the use of short accurate cognitive screening instruments (CSIs) improves case-finding and can aid the development of integrated care pathways [11,12]. Given that CSIs often exhibit poor sensitivity and specificity [12], as well as the well-recognised time–accuracy trade-off, where longer tests are more accurate but time-consuming to administer [13], it is important to look for approaches that balance brevity with sensitivity and specificity.

For researchers, especially within epidemiology, estimates of cognitive impairment are often missing and/or incomparable between studies. For example, the need for more standardised diagnostic criteria has been raised in numerous systematic reviews of MCI examining its prevalence [14,15,16] and incidence [17]. Some reviews have addressed this issue by applying strict inclusion criteria, which has resulted in more harmonised estimates, but also greatly limited the number of studies available for comparison [18,19]. Several efforts have also been made to document the cognitive items or domains (e.g., short-term memory or language) available across large studies of ageing [20,21,22] and identify which are harmonised across different longitudinal studies [23,24]. Many of these population-level longitudinal studies are modelled on the Health and Retirement Study (HRS) in the United States [21] and thus contain similar questions.

The largest of these HRS-based longitudinal studies is the Survey of Health, Ageing and Retirement in Europe (SHARE), which, to date (2004 to 2023), has included 29 countries and 8 main study waves. It focuses on people aged ≥50 years and their partners. While the SHARE includes multiple cognitive assessments, it lacks a cognitive scale combining different cognitive domains to measure impairment over time. This limits the ability of the study and researchers accessing the dataset to (1) track cognitive decline over time in this population, (2) properly examine the epidemiology of MCI and dementia and (3) compare findings with other studies that include established instruments. In this context, the objective of this study is to propose and validate a new, short CSI for use in SHARE. This SHARE cognitive instrument (SHARE-Cog) could then be used to produce harmonised estimates of MCI and dementia in the SHARE, and other similar studies, where it could be replicated.

## 2. Materials and Methods

### 2.1. Sample

This study included participants aged ≥65 years who completed a longitudinal questionnaire in wave 8 (2019/20) of the SHARE [25], which included 19 countries (Austria, Belgium, Croatia, Czech Republic, Denmark, Estonia, France, Germany, Greece, Hungary, Israel, Italy, Luxembourg, Netherlands, Poland, Slovenia, Spain, Sweden and Switzerland). These questionnaires were standardised across countries and were administered via computer-assisted interviews by trained personnel. Further methodological details of the SHARE study have been published elsewhere [26]. Only participants aged ≥65 years who had completed a longitudinal questionnaire were selected for this current analysis as these had a greater number of cognitive tests available. Participants were excluded if they were missing data for the variables of interest, were physically unable to complete the cognitive drawing tests or could not be reliably classified into one of the cognitive diagnostic groups.

### 2.2. Cognitive Tests

#### 2.2.1. SHARE Cognitive Instrument (SHARE-Cog)

The CSI proposed for the SHARE contains three subtests/domains: 10-word registration; verbal fluency, and 10-word recall.

For 10-word registration the interviewee was read a list of 10 words aloud and was asked to immediately recall as many as they could. The task was explained to the participant prior to reading the word list as follows: “Now, I am going to read a list of words from my computer screen. We have purposely made the list long so it will be difficult for anyone to recall all the words. Most people recall just a few. Please listen carefully, as the set of words cannot be repeated. When I have finished, I will ask you to recall aloud as many of the words as you can, in any order. Is this clear?”. One of four word lists was randomly chosen by the computer. The participant only had one attempt.

For verbal fluency the participant was asked to name as many animals as they could within one minute. This test was timed and finished after one minute precisely, with no extensions being applied if the instructions needed to be repeated. The basic instructions would be repeated if the respondent was silent for 15 s: “I want you to tell me all the animals you can think of” and if the respondent appeared to stop early, then they could be encouraged to try to find more words. Any members of the animal kingdom and mythical animals, including different breeds, male, female and infant names within the species, were all accepted. Repetitions, redundancies and proper nouns were not.

The 10-word recall task took place after the verbal fluency test and a serial-7 subtraction task, and the participants were asked to recall as many words as they could from the 10-word registration list. The question was asked as: “A little while ago, the computer read you a list of words, and you repeated the ones you could remember. Please tell me any of the words that you can remember now?”. 

The SHARE-Cog instrument (scoring template) and scoring instructions are presented in the Appendix A.

#### 2.2.2. Cognitive Battery

Additional subtests/domains available in SHARE wave 8 were compiled into a more comprehensive cognitive battery, including six subtests. This acted as a “gold standard” to perform a diagnostic accuracy test. These subtests resembled and often mirrored items included in the Montreal Cognitive Assessment (MoCA) [27] and the same item-scoring approach was applied, leading to a battery with a total score of 16 points (from 0, indicating poor cognition, reflecting an inability to complete the items, to 16 points, suggesting normal cognition).

There were three visuospatial/executive function tasks (5 points), including a clock drawing test, and instructions to copy images of a cube and an infinity loop. The clock drawing test involved drawing the clock face and setting the time to ten past five. Like the MoCA, it was scored with one point for each correct part considering the contour, numbers and hands. For the cube and infinity loop, drawings were scored 1 point each if fully correct. Verbal object naming (3 points) was assessed through verbally describing three objects and asking the participant to name them. The three objects described were scissors from “What do people usually use to cut paper?”, a cactus from “What do you call the kind of prickly plant that grows in the desert?” and a pharmacy from “Where do people usually go to buy medicine?”. Synonyms and names of cacti were acceptable and each correct answer was scored 1 point. Attention/numeracy (4 points) included serial 7s subtraction (3 points) and a counting backwards from 20 task (1 point). The serial 7s subtraction involved counting backwards five times from 100 in 7s, and this task was scored as 3 points for 4/5 correct, 2 points for 2/3 correct, 1 point for 1 correct and 0 if none were correct, as in the MoCA. For the other counting task, participants were asked to count backwards as quickly as they could starting from 20 and were given up to two attempts to complete this task. They got one point if they managed to successfully count backwards from 19 to 10 or from 20 to 11. Orientation (4 points) was also considered, with participants receiving one point for each of the following they got correct: date, month, year and day of the week.

### 2.3. Descriptive Variables

The main descriptive variables included age, sex, education and activities of daily living (ADL). Age was calculated via the year and months of birth and interview, and was divided into 10-year age groups of 65–74, 75–84 and ≥85 years. Education was measured according to the International Standard Classification of Education (ISCED) 1997 [28], and was divided into the following groups: low (0–2, none to lower secondary), medium (3–4, upper secondary and post-secondary non-tertiary) and high (5–6, tertiary) [29]. Other (ambiguous) question responses of “still in school” and “other” were difficult to categorise and were excluded.

ADL difficulties were self-reported such as: “Please tell me if you have any difficulty with these activities because of a physical, mental, emotional or memory problem… exclude any difficulties you expect to last less than three months”. For this study, three cognitively focused instrumental activities of daily living (IADL) were considered: “making telephone calls”; “taking medications”; and “managing money, such as paying bills and keeping track of expenses”. Disability, including physical causes, was assessed using the Global Activity Limitation Indicator (GALI): “For the past six months at least, to what extent have you been limited because of a health problem in activities people usually do? Severely limited; Limited, but not severely or Not limited” [30].

Additional descriptive variables included living alone, employment status, multimorbidity (defined as having at least two comorbidities based on a self-reported list of sixteen conditions), eyesight problems (self-rated as “fair” or “poor”), hearing problems (self-rated as “fair” or “poor”), low self-rated health (rated as “fair” or “poor”), physical frailty (according to the SHARE-FI [31]) and hospitalisation (self-reported as an overnight hospital stay in the last year).

### 2.4. Cognitive Classifications

Participants were divided into the following diagnostic groups: dementia, MCI, SMC and normal cognition using age- and education- specific cut-offs on the cognitive battery (Figure 1, Appendix A), as well as questions on IADL difficulties, subjective memory issues and reports of doctor-diagnosed memory problems. Each of the categories is described below and a detailed overview of the categorisation scheme is provided in Appendix A.

#### 2.4.1. Dementia (D)

Dementia involves cognitive disease that is severe enough to interfere with ADLs [32]. While a clinically confirmed diagnosis of dementia is not possible or available in the SHARE, participants were able to report that “a doctor told [them]… and [they] are either currently being treated for or bothered by… Alzheimer’s disease, dementia, organic brain syndrome, senility or any other serious memory impairment”. If these participants also reported difficulty with cognitively focused/orientated IADLs (telephone calls, taking medications and managing money), they were considered to have dementia. In addition, those reporting difficulties with these cognitively focused IADLs that also scored 2 or more standard deviations (SD) below the mean cognitive battery score by age group and education level (Figure 1) were also considered to have dementia [32].

#### 2.4.2. Mild Cognitive Impairment (MCI)

MCI was determined based on the standard Petersen’s criteria [33]. These participants had objective evidence of cognitive impairment based on testing, a subjective self-reported cognitive complaint, preserved independence in functional abilities (ADLs) and no dementia [33]. The objective cognitive impairment was defined a score between 1 and 2 standard deviations below the mean cognitive battery score by age group and education level [32], as shown in Figure 1. The subjective complaint was taken as a response of “fair” or “poor” to the question: “How would you rate your memory at the present time? Would you say it is excellent, very good, good, fair or poor?”. For independence in functional abilities, the three cognitively focused IADLs considered were telephone calls, taking medications, managing money. In addition, those within the MCI or those in the normal cognitive range with no impairment in IADLs but who reported a doctor provided them with diagnosis of “serious memory impairment” were considered to have MCI.

#### 2.4.3. Subjective Memory Complaints (SMC)

Participants who scored from >1 SD below the mean cognitive battery score and above (Figure 1), and did not meet the criteria for dementia or MCI, were categorised as SMC if they responded “fair” or “poor” to the question: “How would you rate your memory at the present time? Would you say it is excellent, very good, good, fair or poor?”. As those reporting difficulties in IADL were difficult to categorise, they were excluded from this analysis (Appendix A).

#### 2.4.4. Normal Cognition (NC)

Participants who scored from >1 SD below the mean cognitive battery score and above (Figure 1), and did not meet the criteria for dementia, MCI, or SMC, and who did not report difficulties with IADL were considered to have normal cognition (Appendix A).

### 2.5. Statistical Analysis

All analysis was carried out in R (version 4.2.1). The statistical significance of differences between categorical variables by cognitive group was assessed using the Pearson’s Chi-squared test. To determine the optimal weights for each SHARE-Cog subtest, they were included in a multivariate logistic regression model and their regression coefficients were compared. The overall performance of a logistic model (i.e., its statistical fit) was assessed using the pseudo R-squared approach by Estrella ($R_{E}^{2}$), which ranged from 0 to 1, where 1 represents a perfect fit [34,35]. Dominance analysis was carried out to assess the relative importance of each item in the model, measured as the average additional $R_{E}^{2}$ added to the model by including that subtest [36]. The area under the curve (AUC) value of the receiver operating characteristic curves was also used to assess diagnostic accuracy and the 95% confidence intervals (CI) were calculated using DeLong’s method [37]. Covariate-adjusted AUC estimates were used to assess if differences in age, education, sex and country impacted the predictive accuracy according to a frequentist semiparametric approach [38]. For this study, the AUC was interpreted as “excellent” (0.9 to 1), “very good” (0.8 to 0.9), “good” (0.7 to 0.8), “sufficient” (0.6 to 0.7) and “bad” (0.5 to 0.6) [39]. Pearson’s correlation was used to assess the correlation between the three SHARE-Cog items. Internal consistency of the SHARE-Cog was measured using Cronbach’s alpha (α) and the rule-of-thumb cut-offs of unacceptable (<0.5), poor (0.5–0.6), questionable (0.6–0.7), acceptable (0.7–0.8), good (0.8–0.9) and excellent (>0.9) [40].

## 3. Results

### 3.1. Sample Description

This study is a secondary analysis of participants from the SHARE wave 8 and the selection process for inclusion in this study is described in a flow diagram (Figure 2).

The mean age (SD) of the participants was 74.67 (6.61) and 55% were female. Additional descriptive statistics are provided in Table 1 by cognitive group and illustrate that those with better cognitive status also had statistically significant differences in numerous health issues, such as multi-morbidity, frailty and an overnight hospital stay in the last year.

### 3.2. Regression Analysis and SHARE-Cog Weighting

Logistic regression was performed using the three cognitive tests (word registration, verbal fluency and delayed recall) to assess their relative diagnostic importance and to decide how they should be scored within the overall SHARE-Cog instrument (weighting).

#### 3.2.1. Maximum Score for Verbal Fluency

The maximum number of animals named in the verbal fluency task ranged from 0 to 100 in the SHARE. However, based on the $R_{E}^{2}$ value (Appendix A), reducing this maximum animal cut-off as low as 30 had almost no impact on the diagnostic accuracy of the regression model. The value was thus set to 30 for further analysis, equating to an average of one animal named every 2 s. Even where participants named ≥30 animals, the number was capped at 30, with no additional points awarded above this threshold.

#### 3.2.2. Relative Importance of Each Subtest

As presented in Table 2, the regression models including word registration, verbal fluency and word recall achieved good diagnostic accuracy (AUC) and could explain a fair amount of the variability in the outcomes ($R_{E}^{2}$). The dominance analysis found that all three contributed to the performance of the models and are thus all worth including. Word registration contributed more than word recall when distinguishing dementia from MCI, and word recall contributed more when distinguishing MCI from NC.

#### 3.2.3. Scoring of Each Subtest for SHARE-Cog

As presented in Table 2, the regression coefficients illustrate the optimum value per correct word for each subtest in each model. Given the possible variations, a further analysis was carried out, looking at the accuracy (AUC) of all unique weighting ratios (n = 291) created by combinations of 0.5, 1, 2, 3 and 4 points per word, as well as varying the maximum cut-off for animal naming in verbal fluency between 20, 30 and 40 (Appendix A). Based on these results, multiple weightings achieved very similar diagnostic accuracy. Hence, in addition to this, clinical judgement based on the literature was applied to decide on the final weighting used. As verbal fluency and delayed recall are more accurate than registration in diagnostic accuracy test studies of individual patients [41], the weighting of 1 point (for registration), 0.5 points (for verbal fluency, taking a maximum of 30 animals named, and rounding to the nearest whole number) and 2 points (for recall) per word was selected. This gave the SHARE-Cog a total score of 45 points: 10 (registration), 15 (verbal fluency) and 20 (recall) points.

### 3.3. SHARE-Cog Items and Internal Consistency

The 10-word registration subtest ranged between 0 and 10 with a mean number of words of 5.29 (SD: 1.61 and 99th percentile: 9). Verbal fluency ranged between 0 and 100 animals and the mean number of animals was 20.69 (SD = 7.09, 99th percentile = 39). The 10-word recall had a range between 0 and 10 and a mean number of words of 3.9 (SD = 2.05, 99th percentile = 9). Applying the weights and 30 animal limit to verbal fluency, the mean (SD) values of each SHARE-Cog subtests were: 5.29 (1.61) for word registration, 10.14 (3.12) for verbal fluency and 10.58 (3.21) for word recall. The total SHARE-Cog score ranged from 0 to 45 with a mean of 23.24 points (SD = 7.43, 99th percentile = 40). The Pearson’s correlations (r) between the SHARE-Cog subtests were 0.49 for registration and verbal fluency, 0.46 for recall and verbal fluency and 0.73 for registration and recall. The overall internal consistency of SHARE-Cog was “acceptable” (Cronbach Alpha = 0.71).

### 3.4. SHARE-Cog Diagnostic Accuracy

The diagnostic accuracy of SHARE-Cog (Figure 3a) was excellent in its ability to differentiate dementia (from everything else), (AUC = 0.91 and 95% confidence interval (CI): 0.89–0.92) and was very good for separating cognitive impairment (i.e., dementia and MCI) from NC (AUC: 0.83 and 95% CI: 0.82–0.84) and cognitive impairment (dementia and MCI) from SMC and NC (AUC: 0.81 and 95% CI: 0.80–0.82). Similarly, it had good diagnostic accuracy for the other comparisons: MCI versus NC (AUC: 0.79 and 95% CI: 0.77–0.81), MCI versus SMC and NC (AUC: 0.77 and 95% CI: 0.75–0.78) and MCI versus dementia (AUC: 0.76 and 95% CI: 0.72–0.79). Sensitivity analysis was also carried out for different ways of measuring IADLs and excluding the self-reported medical diagnosis question (Appendix A). These confirmed similar diagnostic accuracies.

### 3.5. Adjusting for Covariates

A covariate-adjusted ROC analysis was carried out to assess the impact of age, education, sex and country on the diagnostic accuracy of the SHARE-Cog (Figure 3b, Appendix A). The covariate-adjusted AUC was slightly lower, for example, it provided an AUC of 0.80 (95% CI: 0.79–0.81) for separating cognitive impairment (dementia and MCI) from SMC and NC. Across all the comparisons, differences in age and education level were statistically significantly associated with changes in diagnostics accuracy, with the accuracy being better in those who were older and those who were less educated. For sex, the results were more inconsistent and mostly statistically insignificant. Similarly, few statistically significant differences in diagnostic accuracy were observed between countries when taking Austria as the reference with only Denmark (*p* = 0.029) and Italy (*p* = 0.024) being marginally statistically significant in some comparisons (Appendix A).

## 4. Discussion

This study proposed and validated a new short novel cognitive screen or instrument, SHARE-Cog, which could be used to assess and measure cognitive impairment in the SHARE study. It includes three commonly used cognitive items (subtests): word registration (10 words), verbal fluency (animal naming) and word recall. The weighted scoring of these items in the SHARE-Cog showed good to excellent diagnostic accuracy for cognitive impairment in the SHARE across the spectrum, from identifying MCI to differentiating dementia. Specifically, this analysis showed that SHARE-Cog had excellent diagnostic accuracy for dementia (AUC = 0.91) and very good diagnostic accuracy for cognitive impairment, including MCI (AUC = 0.81). However, for differentiating MCI from normal (objective) performance, including SMC and NC (AUC = 0.77), and for differentiating MCI from dementia (AUC = 0.76) it had good diagnostic accuracy. These results suggest that, based on its relative brevity (estimated to take approximately 2 to 3 min to administer), SHARE-Cog could be a useful screening instrument for studies outside of SHARE and potentially also in clinical practice, though more research is required to confirm this.

These items are also widely available in other longitudinal studies of ageing [23,24], such as The Irish Longitudinal Study of Ageing (TILDA), the English Longitudinal Study of Ageing (ELSA) and the Study on Global Ageing and Adult Health (SAGE), which all share common questions with the University of Michigan’s HRS. Thus, this new tool has the potential to provide a homogeneous approach to measuring cognitive impairment, in a range of studies both in Europe and internationally. Further, while studies have assessed differences in cognitive functioning in the SHARE [42,43,44], to the best of our knowledge, this is the first study to develop and validate a bespoke CSI for this study, as well as the first step toward a harmonised definition of MCI, which the SHARE currently lacks.

The logistic regression analysis found that the optimal scoring of the three subtests depended on which diagnostic comparisons were being made and, upon further analysis, several scores were found to produced very similar predictive accuracy. Relative to the other two assessments, the word registration was weighted higher for differentiating MCI from dementia and word recall was weighted higher when differentiating MCI from NC. It was also found (Appendix A) that reducing the maximum score for verbal fluence from 100 to 30 did not negatively impact the diagnostic accuracy of model. This may be useful, since some studies may not record such large numbers; for example, the Mexican Health and Ageing Study (MHAS) only records 30 [24]. The final SHARE-Cog involved scoring 1 point for each word registered, half a point for each animal in verbal fluency (rounded up to nearest whole number) and 2 points for each word recalled for a total score out of 45.

The dominance analysis assessed the relative importance of each of the three subtests on the performance of the logistic regression models (according to $R_{E}^{2}$). Since the order in which items are added to a model impacts how much value they add this method considers the average contribution across all possible combinations. This analysis found that all three items contributed positively to the model and suggested that word registration was statistically capturing something in addition to delayed recall. This may be due to the fact that it involved 10 words, which was a large number. Previous research has also suggested that, in immediate recall learning tasks, such as a short form of the California Verbal Learning Test, may necessitate the use of networks beyond traditional episodic memory, such as semantic processing [45]. Despite this, other CSIs vary on whether or not they score immediate recall, for example, both the Mini-Mental State Examination (MMSE) [46] and the Quick Mild Cognitive Impairment (Q*mci*) screening [47] score it, but the MoCA [27] does not. This needs to be considered carefully in future studies, where scoring it may improve the diagnostic accuracy or reduce the number of additional subtests.

The covariate adjustment found that the diagnostic accuracy was consistently affected by age and education, where the screening was more accurate for those who were older and those with lower education levels. Hence, SHARE-Cog may be a particularly useful tool for those with low literacy or education. This is supported by research that found that tasks involving “reading, writing, arithmetic, drawing, praxis, visuospatial and visuoconstructive skills have a greater educational bias than naming, orientation, or memory” [48]. While they were excluded from the current study, SHARE-Cog can also be administered to those with physical limitations that affect their ability to complete drawing and written tasks, since all three tasks in SHARE-Cog are verbal. SHARE-Cog may be slightly worse at distinguishing MCI from normal cognition in women (*p* = 0.02), although this may just be a chance finding, because, while statistically significant, the difference was marginal from a clinical perspective. Similarly, for countries there were very few statistically significant differences suggesting that country has little to no impact on its diagnostic accuracy within the European context. 

The strengths of this study include the large sample size and that it was a population-based cohort. Given that CSIs are mostly assessed in clinical settings, there is a need for more population-level studies to assess their utility within community settings. There are several limitations, however, relating to the accuracy of the dementia and MCI diagnostic categories. Firstly, the responses were self-reported, potentially introducing bias. For example, the self-reported “doctor-diagnosed serious memory problem” question likely included a range of conditions. This could have resulted in misclassification. For this study, attempts were made to separate it into dementia and MCI. The SMC and ADL questions were also subjective “tick box” questions that would lack the accuracy and rigour of a formal clinical investigation. Similarly, the reasons for ADL limitations were not available and could have been related to cognition, physical or emotional issues. We attempted to isolate more cognitively focused tasks, but this approach may have introduced bias. However, sub-analysis assessed the impact of different approaches to measuring these and did not find that varying them had little impact on the diagnostic accuracy of the SHARE-Cog for overall cognitive impairment (Appendix A). In addition, many SHARE participants were excluded based on their diagnostic group being too unclear/contradictory to quantify clearly. 

It is also unclear if the results of this study are generalisable to all waves of SHARE, other longitudinal studies or clinical practice i.e., spectrum bias; hence, more research is needed to evaluate and externally validate SHARE-Cog. Given that the participants had received a previous SHARE interview, there may have been a small risk of practice effects, although this is unlikely, since there were, on average, 2–3 years between interviews. Even in this context, four random lists of words were used for word registration and recall, minimising learning effects. Verbal fluency was only for animals, but this is not as prone to learning/practice effects as other subtests. A previous study suggested that, in those with MCI, practice effects can last up to one year, but are more marked for word registration and recall than other cognitive domains, such as verbal fluency [49]. Most of the subtests used in the broader cognitive battery were only introduced in the current wave. 

Our analysis focused on generating a CSI (SHARE-Cog), and further research could look at generating a more complete risk-prediction model including both cognitive measures and patient characteristics (such as age, sex and education and likely risk factors, such as hypertension, etc.) [50].

## 5. Conclusions

In conclusion, SHARE-Cog performed well, demonstrating good to excellent diagnostic accuracy across the spectrum of cognitive impairment in this validation study. Despite being only composed of three subtests, it could be used in analyses of the SHARE and numerous other longitudinal studies of ageing to better understand the epidemiology of cognitive impairment. This could aid in harmonisation efforts quantifying the prevalence of syndromes such as MCI and cognitive frailty, which currently are extremely heterogeneous across studies, thus making epidemiological comparisons challenging.

## Figures and Tables

**Figure 1 ijerph-20-06869-f001:**
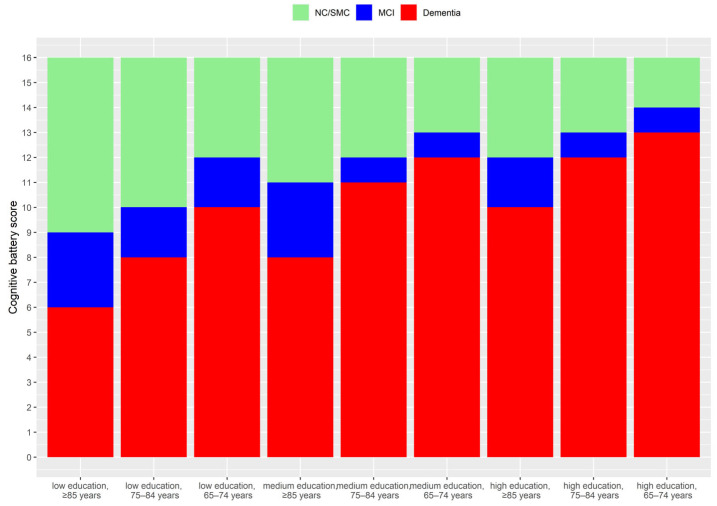
Age and education-specific cut-offs applied for the identification of an objective cognitive deficit on testing. Cut-offs were defined based on being 1 or 2 standard deviations (SD) below the mean cognitive score.

**Figure 2 ijerph-20-06869-f002:**
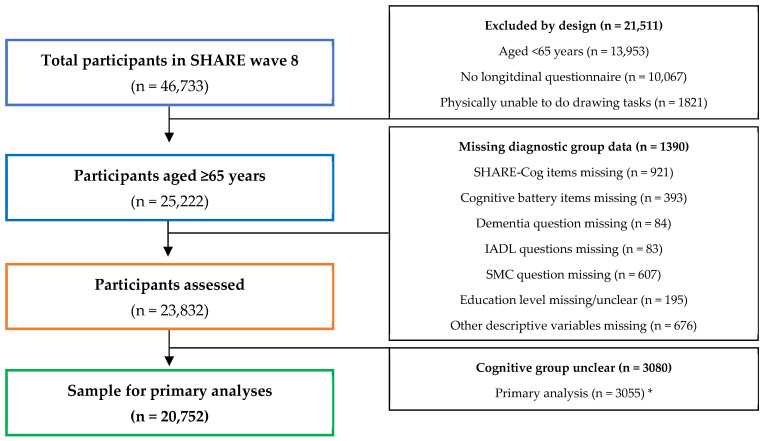
Flow diagram illustrating participant selection and reasons for exclusion from this secondary analysis. * Sensitivity analyses (presented in the Appendix A) excluded different numbers due to the cognitive groups being unclear and details on these are provided in Appendix A.

**Figure 3 ijerph-20-06869-f003:**
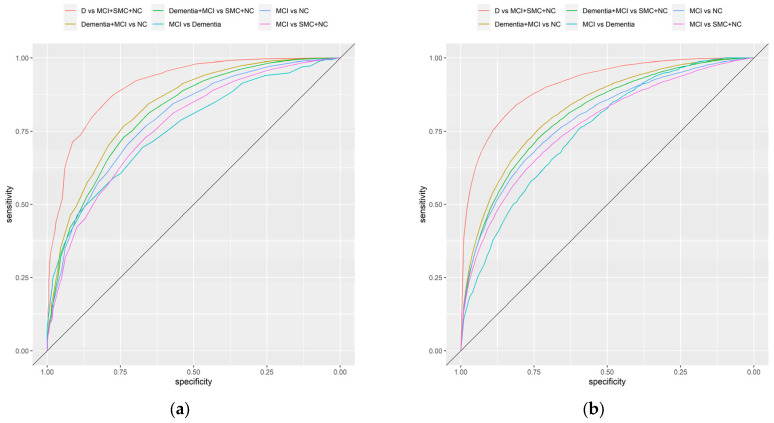
Receiver operating characteristic curves illustrating the diagnostic accuracy of the SHARE-Cog including: (**a**) unadjusted curves and (**b**) covariate-adjusted (age, sex, education and country) curves.

**Table 1 ijerph-20-06869-t001:** Descriptive characteristics of the participants by diagnostic category: dementia, mild cognitive impairment (MCI), subjective memory complaints (SMCs) and normal cognition (NC).

Descriptive Variable	Total (n = 20,752)	Dementia (n = 335)	MCI (n = 778)	SMC (n = 4957)	NC (n = 14,682)	Difference (*p*-Value)
Age: 65–74 years	56%	27%	49%	46%	61%	<0.001
Age: 75–84 years	35%	47%	37%	42%	33%	
Age: ≥85 years	8%	26%	14%	12%	6%	
Female	55%	59%	58%	54%	55%	0.046
Education level: low	35%	50%	49%	39%	33%	<0.001
Education level: medium	40%	33%	31%	39%	41%	
Education level: high	24%	17%	19%	21%	26%	
GALI ^1^: activities not limited	51%	12%	33%	40%	57%	<0.001
GALI ^1^: limited but not severely	35%	31%	41%	40%	32%	
GALI ^1^: severely limited	14%	57%	26%	19%	11%	
Lives alone	29%	37%	36%	32%	27%	<0.001
Employed	4%	0%	2%	4%	5%	<0.001
Multimorbidity	53%	72%	65%	59%	50%	<0.001
Eyesight problems	18%	43%	33%	27%	14%	<0.001
Hearing problems	22%	42%	37%	36%	15%	<0.001
Low self-rated health	37%	80%	64%	57%	28%	<0.001
Physical frailty	7%	43%	15%	9%	5%	<0.001
Hospitalisation ^2^	16%	34%	20%	19%	15%	<0.001

^1^ GALI (Global Activity Limitation Indicator): activities limited by health for at least 6 weeks. ^2^ Overnight hospital admission (stay) in the last year.

**Table 2 ijerph-20-06869-t002:** Regression modelling (unadjusted) of the three SHARE-Cog items for different diagnostic comparisons including the regression coefficients for the best fit, the overall diagnostic performance of the model and the relative importance of each item in regards to the model performance (dominance analysis).

Diagnostic Comparisons	Regression Coefficients				Scoring *	Model Performance		Dominance Analysis **		
Groups Being Compared	(Intercept)	Word Registration	Verbal Fluency	Delayed Recall	Rounded Score per Word	AUC	$R_{E}^{2}$	Word Registration	Verbal Fluency	Delayed Recall
(D + MCI) vs. (SMC + NC)	0.877	−0.218	−0.109	−0.296	2:1:3	0.811	0.141	0.025	0.032	0.028
(D + MCI) vs. (NC)	1.524	−0.244	−0.109	−0.347	2:1:3	0.831	0.199	0.038	0.044	0.045
MCI vs. D	−1.722	0.322	0.106	0.054	6:2:1	0.765	0.212	0.076	0.086	0.036
MCI vs. (SMC + NC)	−0.228	−0.12	−0.085	−0.288	1:1:3	0.767	0.053	0.01	0.014	0.015
MCI vs. NC	0.511	−0.158	−0.088	−0.334	2:1:4	0.791	0.091	0.017	0.021	0.026
D vs. (MCIc+ SMC + NC)	1.312	−0.444	−0.186	−0.353	2:1:2	0.908	0.211	0.022	0.027	0.017

D = dementia, MCI = mild cognitive impairment, SMC = subjective memory complaint, NC = normal cognition. AUC: Area under the curve from ROC curves. * The scoring here is the rounded ratio between the three regression coefficients. ** The dominance analysis is the average addition of each variable to the $R_{E}^{2}$ value of the model. In other words the relative importance of each variable.

## Data Availability

Data may be accessed through becoming a registered user with the Survey of Health, Ageing and Retirement in Europe (via www.share-project.org) (accessed on 7 September 2023).

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
