# Peer review of "Validation of a Harmonised, Three-Item Cognitive Screening Instrument for the Survey of Health, Ageing and Retirement in Europe (SHARE-Cog)"

_ijerph, 2023, doi:10.3390/ijerph20196869_

Round 1

Reviewer 1 Report

The manuscript focuses on the validation of the three-item cognitive screening instrument for the Survey of Health, Ageing and Retirement in Europe (SHARE-Cog), which is a very useful method; researchers and health professionals should be increasingly interested in the results found and reported.

I find the manuscript to be well written and grounded; the analysis is accurate and well documented.

It is clear that the authors have focused on the diagnostic accuracy of the Share-Cog measurement tool; however, it would be extremely useful to see, for example, how this accuracy varies across countries. In addition, the correlation between the three items, or the internal consistency of the three-item scale, might also be of interest, not to mention the analysis of the measurement invariance of the instrument by countries. I am aware that not everything fits in a paper, but it might be worthwhile to emphasize these aspects, either in the limitation or in future studies.

Congratulations to the authors for a very valuable and thorough work, I hope that my comments will help to improve the manuscript.

Reviewer 2 Report

Thank you for the opportunity to review this manuscript. This manuscript examined how dementia and mild cognitive impairment can be distinguished, using the SHARE retrospectively. 

As the authors state, the major strength in this manuscript would be the number of patients enrolled in the study. However, under this circumstance, the reviewer would have preferred if the authors could divide the patients into two groups, possibly arranging its background, and check to see whether the protocol would be able to classify the patients into each cognitive group using the algorithm derived from the other group. This may result in a more robust classification than comparing the AUCs. 

The authors could consider bringing the final paragraph of the introduction (where they explain the SHARE) in the materials and methods, possibly in 2.1. The current style seems to introduce SHARE quite suddenly. 

The authors could consider rearranging their abstract as well; the connections between several sentences could be improved to provide context to the abstract.  

Reviewer 3 Report

Identifying dementia and mild cognitive impairment at the population level is a critical step to advance the field of dementia research.  The authors have proposed a fairly simple cognitive assessment to examine MCI and dementia with lowered burden to participants in the SHARE cohort (n~20,000 for this analysis).  There are a couple of points the authors should clarify for publication.

Naming 100 animals in 1 minute seems unrealistic.  Were any values excluded for non-plausibility?

Are all of the regression models in Table 2 unadjusted?

Was wave 8 the first time the cognitive battery was administered?  Could there have been any learning effects?

It was not clear that sensitivity analyses were done to examine robustness of the models across various sub-groups? Were there any effects due to country or education?
